# The Effects of *Spirulina maxima* Extract on Memory Improvement in Those with Mild Cognitive Impairment: A Randomized, Double-Blind, Placebo-Controlled Clinical Trial

**DOI:** 10.3390/nu14183714

**Published:** 2022-09-09

**Authors:** Woon-Yong Choi, Won-Kyu Lee, Tae-Ho Kim, Yong-Kyun Ryu, Areumi Park, Yeon-Ji Lee, Soo-Jin Heo, Chulhong Oh, Young-Chul Chung, Do-Hyung Kang

**Affiliations:** 1Jeju Marine Research Center, Korea Institute of Ocean Science and Technology (KIOST), Jeju 63349, Korea; 2Department of Ocean Science, University of Science and Technology (UST), Jeju 63349, Korea; 3Department of Psychiatry, Jeonbuk National University Medical School, Jeonju 54907, Korea

**Keywords:** microalgae, cyanobacterium, *Spirulina maxima*, clinical trial, memory improvement, Alzheimer’s disease, functional food

## Abstract

*Spirulina maxima* is a marine microalga that has been promoted worldwide as a super food. This study was conducted to evaluate its ability to improve memory in the older adults using *Spirulina maxima* 70% ethanol extract (SM70EE). This randomized, double-blind, placebo-controlled clinical trial comprised 80 volunteers recruited from Jeonbuk National University Hospital in Jeonju, Republic of Korea, who were randomly assigned to two groups. The participants received either 1 g/day of SM70EE or a placebo without otherwise changing their diet or physical activity. The participants were examined at baseline and after a 12-week interval to determine whether there were changes in their results for visual learning, visual working memory, and verbal learning tests from the Korean version of the Montreal Cognitive Assessment, brain-derived neurotrophic factor and beta-amyloid levels, and total antioxidant capacity. Compared to the placebo group, the treatment group showed a significant improvement in visual learning and visual working memory test results and enhanced vocabulary. SM70EE use was shown to improve memory, with no adverse effects. Its efficacy in alleviating Alzheimer’s disease symptoms was verified for the first time through this clinical trial. SM70EE could play a role in the management of patients with dementia. This trial is registered with registration number of clinical research information service (CRIS: KCT0006161).

## 1. Introduction

An increase in the elderly population and low birth rates are a worldwide phenomenon, and this aging trend is shown to be closely related to the prevalence of memory impairment [1,2]. While memory impairment cannot yet be completely prevented, the importance of early intervention to delay onset by slowing down the condition or improving the symptoms has been emphasized [3]. Alzheimer’s disease (AD), which involves the most severe level of memory impairment, is mostly irreversible, thus it is critical to prevent deterioration of cognitive function [3,4]. *Spirulina* is a microalga known to be high in nutrients and is considered one of the most perfect dietary supplements for humans because it contains many bioactive substances. As a result, *Spirulina* has been primarily studied in biological activities such as obesity and high blood pressure [5,6]. *Spirulina maxima* 70% ethanol extract (SM70EE) has been administered to improve memory impairment, with previous studies with scopolamine and amyloid-β (Aβ)-induced memory impairment models showing a relation between SM70EE administration and increased antioxidant effects [7,8,9]. This led to a significant improvement in cognitive outcomes in animal behavior experiments, such as the Morris water maze and passive avoidance experiments [7,8,9]. The development of new approaches and applications to promote memory improvement is critical for the prevention and delay of AD, particularly in its early stages [3,4]. A mild cognitive impairment (MCI) assessment can be used as an index to determine early potential memory impairment and indicate whether improvement is occurring among patients in the pre-AD stage following specific interventions [10,11]. When comparing the differences between individuals with MCI and those without MCI, MCI appears to be a transitional phase between those with AD and those without AD, affecting immediate memory, working memory, continuous attention, and selective attention [12,13]. MCI is known to particularly affect the visuospatial working memory before the verbal working memory [14,15]. Between 1% and 2% of healthy individuals develop AD each year, and 10–15% of those with MCI develop AD [16]. For this reason, improving the memory function of those with MCI is likely to be helpful in preventing or delaying AD because MCI is an index indicating an individual’s AD risk status.

In this study, the Korean version of the Montreal Cognitive Assessment (MoCA-K) and a computerized neurocognitive function test (CNT) were used to check the effects of SM70EE on memory function in those with MCI, in accordance with previous studies [17,18]. The characteristics of our study were analyzed in terms of the sample population, sample size, eligibility criteria, the proportion of the population for which data were analyzed, mean age with standard deviation, presence of a control group, timing of assessment, and education level in relation to the MoCA-K score. To identify the clinical effects of SM70EE on those with MCI, prototype capsules of SM70EE were taken daily and safety assessments involving validity evaluations in terms of the MoCA-K, CNTs, brain-derived neurotrophic factor (BDNF) and Aβ levels, total antioxidant capacity (TAC), and diagnostic tests were conducted 12 weeks later. The SM70EE is effective against memory damage and inhibited memory loss caused by β-amyloid (Aβ) deposition, as shown in our previous studies performed on animals [8,9,17]. However, there are no reports on the positive effects of SM70EE on human cognitive function. Therefore, this study aimed to report the effects of SM70EE on cognitive function in older adults (aged over 60 years) with memory impairment.

## 2. Materials and Methods

### 2.1. Study Design and Population

This clinical study was a randomized, double-blind, placebo-controlled clinical trial that was designed and conducted over a duration of 12 weeks to evaluate the memory improvement effects of SM70EE use compared to that of a placebo [19,20]. The number of individuals determined as necessary to participate in the trial (patients with MCI) to ensure validity was 80, allowing for a possible 20% dropout rate that would leave 64 individuals. After completing the consent form to participate in the study, screening tests were performed on 80 volunteers recruited from Jeonbuk National University Hospital in Jeonju, Republic of Korea, who were divided into an SM70EE group and a placebo group. They were found to have an average educational background of 10.31 ± 3.75 years (SM70EE group 10.58 ± 3.66 years, placebo group 10.05 ± 3.86 years), and an average age of 68.26 ± 4.68 years (SM70EE group 67.68 ± 4.43 years, placebo group 68.85 ± 4.89 years). Participants with a history of surgery, severe health problems including hypertension, clinical depression, diabetes mellitus, and thyroid diseases, those on any medication or supplements that might affect metabolism, those with a history of allergy to medicinal plant extracts or placebo products, and pregnant or lactating women were excluded from the study cohort. Participants were encouraged to follow healthy lifestyle habits and were advised to not make any significant changes to their diet and routine physical activity during the 12 weeks of the study protocol. This interventional study was registered as Protocol no. KIOST-CF-SM1 using the protocol registration system and was approved by the medical ethics committee of the institutional review board of Jeonbuk National University Hospital (CRIS: KCT0006161). All participants signed an approved, written consent form at the initiation of the study [5,21].

### 2.2. Randomization and Blinding

The 80 eligible participants were randomized into two equal-sized groups of 40 individuals each through an online randomization program (http://www.randomization.com/, accessed on 21 April 2017). The investigators and participants were blinded to the treatment regimens (double-blinded). The SM70EE and placebo samples were coded by a coinvestigator who was not involved in the study. The eligible study participants were randomly allocated to consume either a 1 g sample of an active SM70EE product or a placebo product, three times a day for 12 weeks [5,21].

### 2.3. Preparation of SM70EE and the Placebo

The Korea Institute of Ocean Science and Technology (KIOST) (Jeju, Republic of Korea) provided the SM70EE using a two-step method, as described in our previous studies [7,8,9]. In brief, *S. maxima* was cultured in a medium within a 6-ton-scaled open raceway pond at the KIOST Jeju Marine Research Center to ensure the stability and safety of the chemical cell components according to the rules of the Korean Ministry of Food and Drug Safety under serial batch conditions [22]. *S. maxima* samples were extracted with 70% ethanol at a ratio of 1:10 (*w*/*v*) through ultrasonic wave at a frequency of 40 kHz and room temperature for 8 h. Thereafter, the SM70EE pretreated was then extraction at 65 °C for 4 h until the end of the process. In our previous study, both the chlorophyll-a treatment group and the SM70EE treatment group showed neuroprotective activity [9]. Chlorophyll in SM70EE was a major component and contained a concentration of 15.68 ± 0.76 mg/g. Since chlorophyll-a had the efficacy in a concentration-dependent experiments, chlorophyll in SM70EE as a mixture was used as an indicator in the clinical trials [7,8,9]. The placebo was prepared by dissolving gardenia pigments (0.5% *w*/*w*) and caramel color (0.5% *w*/*w*) into drinking water, and was identical to the SM70EE product in appearance and flavor [5,21].

### 2.4. Clinical Trial

This 12-week, randomized, double-blind, placebo-controlled clinical trial was performed to evaluate the efficacy and safety of SM70EE for improving memory function. The following factors were evaluated by comparing the effects of SM70EE with that of the placebo. First, the effectiveness of SM70EE in improving memory function, as evaluated using visual learning, visual working memory, and verbal learning tests, was compared with that following placebo intake. Second, MoCA-K results, BDNF and Aβ levels, and TAC were assessed to evaluate the efficacy and safety of SM70EE for improving memory function compared to that of the placebo [23,24,25]. The inclusion criteria for the participants were as follows: (1) age 60 years or older at the time of the screening test, (2) those who scored 25−28 points on the Korean mini-mental status examination (K-MMSE), and (3) those who agreed to voluntarily participate and abide by the study requirements as set out in writing after listening to the detailed explanations and fully understanding the processes. The participants were first visited within 4 weeks from the screening visit and enrolled in the clinical test after confirming their suitability. Subsequently, the participants were randomly assigned into either the SM70EE or placebo group to complete the baseline evaluation until the first visit [23,24,25]. After group assignment, the participants consumed the relevant SM70EE or placebo samples three times a day (capsule whole weight 2.4 g/day) after breakfast/lunch/dinner, for 12 weeks (SM70EE group: 1 g/day of SM70EE; placebo group: 1 g/day without SM70EE) [8,9,17,26]. Subsequently, every 6 weeks, the participants visited the Functional Food Clinical Trial Support Center at Cheonbuk National University Hospital and underwent tests related to memory improvement, vital signs, drug administration, changes in medical conditions, adverse reactions, and tests specified in the clinical test plan. The research investigator conducted the necessary follow-up observations of the participants after the final intake of the test product or after an early termination visit to derive the results (Figure 1) [23,24,25].

### 2.5. Measurement of Biomarkers

The evaluation of biomarkers in plasma was performed in an ethylenediaminetetraacetic acid (EDTA) tube, and the tube was reversed 10 times immediately after the sample was taken and mixed. The sample was centrifuged at 3000 rpm for 10 min, with 1 mL of supernatant transferred to a 1.5 mL microtube. The sample was then stored in a deep freezer (−70 °C). Aβ was measured in the plasma samples using an enzyme-linked immunosorbent assay (ELISA) using the Human Aβ Assay Kit (Immuno-Biological Laboratories, Takasaki-shi, Japan), which is a solid-phase sandwich ELISA test [19]. BDNF was measured in plasma samples using a BNDF rapid ELISA kit (Biosensis, Thebarton, Australia; mature BDNF; human, rat, mouse) [27]. Total antioxidant capacity (TAC) was measured using the ferric reducing antioxidant power method developed by Benzie and Strain [28].

### 2.6. Statistical Analysis

Efficacy was evaluated using an independent *t*-test for differences in changes in primary validation indicators (visual learning, visual working memory, and verbal learning tests) and secondary validation indicators (MoCA-K, BDNF, Aβ, and TAC) between the test and control groups, before and after 12 weeks of product consumption. Additionally, a paired *t*-test was used to analyze the difference in changes in the results of the 7 tests in the test and control groups, before and after 12 weeks of product consumption. Non-homogeneous demographic information factors were corrected using covariates and analysis of covariance. The safety evaluation was summarized in terms of the frequency and percentage of all adverse reactions that occurred for all participants and the participants in each group during the clinical study period. The incidence of abnormal findings between the two groups was analyzed using chi-square or Fisher’s exact tests. Diagnostic laboratory medical tests and vital signs were compared among those in the test group and between those in the test and control groups. For comparison within the test group, the average change was analyzed using a paired *t*-test. For comparison between the two groups, the average amount of change was compared and analyzed using an independent *t*-test. The program used for the analysis was SAS version 9.2 (SAS Institute, Cary, NC, USA), SPSS version 20 (IBM Corp., Armonk, NY, USA), GraphPad Prism version 8.1.1 (GraphPad Software Inc., San Diego, CA, USA), and the statistical significance level was set at *p* < 0.05.

## 3. Results

### 3.1. Demographic Information and Compliance Assessment

A total of 80 participants (40 in the SM70EE group and 40 in the placebo group) participated in this clinical test; the results of the demographic information analysis are shown in Table 1.

Of the study participants, 59 were women (28 in the SM70EE group, 31 in the placebo group) and 21 were men (12 in the SM70EE group, 9 in the placebo group), with an average educational background of 10.31 ± 3.75 years (10.58 ± 3.66 years in the SM70EE group, 10.05 ± 3.86 years in the placebo group) and an average age of 68.26 ± 4.68 years (67.68 ± 4.43 years in the SM70EE group, 68.85 ± 4.89 years in the placebo group), with no statistically significant difference between the groups (*p* > 0.05).

The mean K-MMSE score was 26.38 ± 1.06, with no significant difference between the two groups, and no other variables were statistically significant between the groups (*p* > 0.05). To summarize these results, no significant differences in variables were found between the groups, indicating that the randomization process had been effective. The compliance of the participants was measured, and the results are shown in Table 2. The total number of products consumed by one person amounted to 252.74 ± 7.30 (capsules) on average, with no differences between the SM70EE and placebo groups (*p* = 0.477), and no statistically significant difference in each group (*p* = 0.715). Product compliance was not significantly different between the two groups (92.41 ± 8.15 in the SM70EE group and 93.43 ± 6.27 in the placebo group; *p* = 0.545).

### 3.2. Comparison of SM70EE and Placebo Groups during the 12-Week Trial

Findings in terms of the variables over 12 weeks for the SM70EE and placebo groups are shown in Table 3 and Table 4. The analysis did not include one person (SM70EE group) who was excluded due to lack of compliance, four people (one in the SM70EE group and three in the placebo group) who were excluded as they were found to be using prohibited drugs, and one person (SM70EE group) who dropped out of the study. Table 3 shows the results of the CNTs for visual memory, visual working memory, and verbal memory, when comparing the amount of change before and after 12 weeks of SM70EE intake. There were statistically significant differences between the groups after 12 weeks regarding their visual learning (*p* = 0.007) and visual working memory (*p* = 0.024) test results. Other variables did not differ significantly between the two groups (*p* > 0.05). Table 4 shows the comparison of changes before and after 12 weeks of SM70EE intake in MoCA-K, except for certain anomalies in relation to the secondary validation indicator. There was a statistically significant difference between the two groups in vocabulary (*p* = 0.043). Other variables did not differ significantly between the two groups (*p* > 0.05). Table 5 shows that there was no statistically significant difference in changes in secondary validation indicators, namely, BDNF and Aβ levels between the two groups before and after SM70EE intake (*p* > 0.05). As a matter of interest, although the increase in the TAC was not significant (*p* > 0.066), the efficacy increased to a level that will need to be verified once again through further studies. The findings of this study are important, as measuring the high levels of reactive oxygen species in cellular degenerative processes is easy using human cells, whereas measuring TAC in the blood as a nutrient carrier for cells is difficult. Summarizing the above results, the significant changes resulting from the administration of SM70EE to patients with MCI were improvements in visual learning and visual working memory test results, vocabulary, and TAC (Figure 2).

### 3.3. Safety Issues and Adverse Events

The frequency of occurrence of adverse events was measured to evaluate the stability of SM70EE, and the results are shown in Table 6. One participant did not comply with the study expectations (SM70EE group). However, no significant difference was observed in the occurrence of adverse events between the two groups (*p* > 0.05), and there was no causal relationship with SM70EE consumption. Diagnostic laboratory tests (blood tests, urine tests, and other relevant tests) were undertaken, and the results are shown in Table 7, with no statistically significant differences found between the two groups (*p* > 0.05) and no clinically significant abnormalities or changes indicated in relation to consuming SM70EE. The results of the evaluation of vital signs in the SM70EE and placebo groups are shown in Table 8, with no statistically significant difference found between the two groups (*p* > 0.05). Finally, body weight, body mass index, and metabolic equivalent of task test results of the SM70EE and placebo groups are shown in Table 9 and Table 10, with no statistically significant differences observed between the two groups (*p* > 0.05).

## 4. Discussion

This randomized, double-blind, placebo-controlled clinical 12-week trial was conducted to assess the effectiveness and safety of SM70EE use for memory improvement. Clinical studies on *Spirulina* strains have been conducted in relation to their use in obesity and high blood pressure; however, no results of clinical trials of its effects on memory improvement have been reported [5,6]. Preclinical studies for clinical trials of SM70EE have shown an enhanced cranial nerve protection effect in the hippocampal neuronal cell line (HT22 cell) of mice in in vitro tests [7,8,9]. In addition, an increase in mitochondrial membrane potential and a decrease in reactive oxygen species (ROS) have been found to play an important role in the inactivation of the mitochondrial ROS scavenging system [7,8,9]. If these mitochondrial ROS scavenging systems are interrupted or the electron transport chain is broken, ROS byproducts are released [29,30]. Excessive ROS is known to cause aging due to protein damage in cells as well as in mitochondria [29,30]. The memory improvement effect of SM70EE has been reported to be related to the antioxidant action caused by chlorophyll and some extracted components of *S. maxima* [9]. It would appear that the in vitro activation mechanism involved in SM70EE improves neuron synapse function through inhibiting the production of ATP in the mitochondria in response to nerve cell damage in the cerebral cortex [31,32]. In addition, SM70EE has been identified as having long-term and short-term memory improvement effects in animal experimental models. SM70EE also leads to improved memory function due to increased protein expression of BDNF, cyclic adenosine 3′,5′-monophosphate response element binding protein (CREB), and extracellular signal-regulated kinases (ERK), and decreased acetylcholinesterase activity [7,8,9], with significant memory improvement shown in cumulative Aβ in mouse models. Clinical trials have been proposed using SM70EE, as it has been considered a possible means to improve AD [7,8]. Plasma biomarker measurements identified in the clinical tests showed no statistically significant differences in relation to MoCA-K results and BDNF and Aβ (Aβ40/Aβ42) levels. These plasma biomarkers have been reported to have varying effects depending on individual health or age, which may lead to different results from animal experiments [31,32,33,34]. The TAC showed no clearly significant difference, but there was a higher significance compared to other biomarkers. As the average age of the participants (68.26 ± 4.68 years) was high, there was no clear difference in the TAC results obtained after consuming SM70EE, which aligned with the results reported in clinical studies on immunity enhancement using caterpillar fungus (*Cordyceps militaris*) extract, in which differences in initial antioxidant effects and immunity due to sex, average age, and other various factors were assessed [31,32,35]. In addition, plasma BDNF has been measured in the later phase of the AD spectrum in previous studies [36,37,38] and showed no significant difference; therefore, it was not measured in this study. The plasma Aβ concentration in patients with MCI has been reported to be similar to that in healthy individuals, with plasma Aβ-40 reportedly being approximately 270−280 pg/mL and plasma Aβ-42 being approximately 5−20 pg/mL, indicating that the Aβ concentration was very similar to the results of this study [30,33,35,36]. However, in patients with AD, plasma Aβ-40 has been reported to be 240−250 pg/mL and Aβ-42 to be 5−15 pg/mL, which are lower levels than those of healthy individuals or those with MCI [33,37,39,40]. Therefore, this study showed that the consumption of SM70EE did not significantly change Aβ accumulation in the plasma of patients with MCI [33,37,39,40]. An individual’s state of health or age could be an important factor affecting biomarker levels in clinical trials.

However, when comparing changes before and after consuming SM70EE through CNTs for the primary indicator validity assessment, we found statistically significant differences in visual learning and visual working memory tests. In addition, the results from the MoCA-K in terms of vocabulary showed a statistically significant difference between the two groups in the secondary indicator validity assessment. In particular, SM70EE consumption was found to be safe for the human body because no clinically significant adverse reactions or physical changes were observed. These findings were consistent with recent studies on SM70EE concerning memory improvement and SM70EE stability [7,8,9]. Validity evaluation results showed significant increases in MoCA-K vocabulary and improvement in CNT visual assessment results, which demonstrated that there was an improvement in the memory function of those with MCI [13,14,15,16]. The visual working memory is reportedly more damaged than the verbal working memory in patients with AD. However, constraining the process whereby short-term language memory is converted into long-term language memory resulted in memory impairment [13,14,15,16]. Based on these findings, the improvements found in the visual learning and visual working memory tests following the use of SM70EE indicated that the improvement of memory function was most likely a definite effect. It has also been shown that patients with AD have lower visual working memory test results than healthy individuals [13,14,15,16]. In particular, early symptoms of MCI have reportedly been associated with reduced storage of visuospatial working memory, due to shrinking of the hippocampus, which is responsible for visual and spatial memory [13,14,15,16]. Therefore, SM70EE use in this study was considered to have improved the effectiveness of visual and spatial memory in patients with MCI and to have inhibited their progression to AD. Patients with early AD, such as those with MCI, develop early atrophy of the fusiform gyrus, that is responsible for visual-grapheme-phoneme conversion near the hippocampus, due to atrophy and damage to the hippocampus responsible for visuospatial working memory [41,42,43,44]. Specifically, when a pre-linguistic visual stimulus is received, it undergoes a conversion process in which the visual stimulus is phonetically encoded. However, issues arising in this chain reaction can negatively affect the process of phonetic conversion from visuospatial stimuli in patients with MCI or AD more than it can in healthy individuals, particularly in terms of vocabulary [41,42,43,44]. Therefore, the significant increase in vocabulary found in this study as evaluated using the MoCA-K would appear to be due to the effects of visual memory improvement and related synergies. Overall, SM70EE use showed an enhanced memory improvement effect accompanied by an increased vocabulary, due to improvements in the visual learning memory and visual working memory [43,44]. Therefore, continuous consumption of SM70EE by individuals in the MCI phase of early AD could lead to improved memory function through improved visual memory and vocabulary.

Our results indicated no differences in the variables between the groups, and the randomization process was effective. The 12-week supplementation with SM70EE was shown to be valid and safe. Therefore, the significant findings of this study suggest that SM70EE supplementation in patients with MCI with memory disorders is safe and can play a role in the prevention and management of AD.

## 5. Conclusions

This study was the first clinical trial to evaluate the efficacy and safety of *S. maxima* extract (SM70EE) for improving memory function and was conducted using a 12-week randomized, double-blind test, and place-controlled clinical trial. SM70EE intake effects on the CNT results, which is the primary validation indicator, showed statistically significant differences between the visual leading test and the visual working memory test. Moreover, as confirmed by the secondary validation per the MoCA-K, the vocabulary results showed a statistically significant difference between groups. SM70EE ingestion was determined to be safe for the human body because no clinically significant adverse reactions or physical changes were observed. Therefore, the improvement of memory function activity by SM70EE ingestion was first confirmed in clinical tests, and patients with early AD such as MCI confirmed that continuous intake of SM70EE could show improvement in memory function through improved visual memory and vocabulary.

## Figures and Tables

**Figure 1 nutrients-14-03714-f001:**
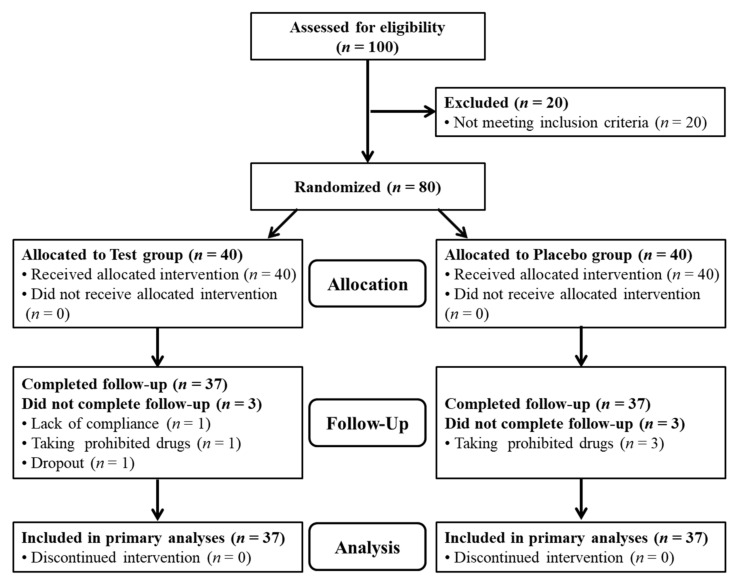
Trial profile.

**Figure 2 nutrients-14-03714-f002:**
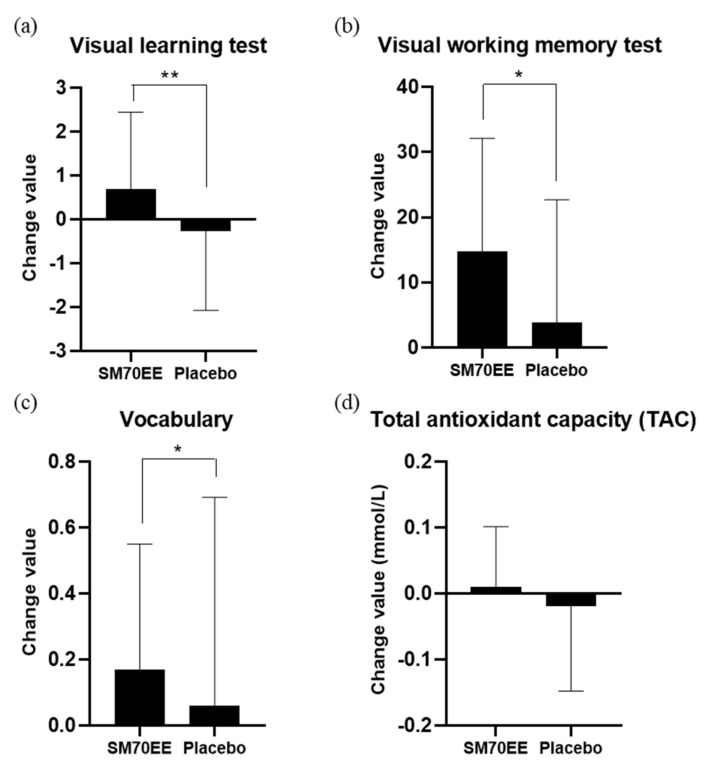
Changes of values in (**a**) visual learning test, (**b**) visual working memory test, (**c**) vocabulary (MoCA-K) and (**d**) total antioxidant capacity (TAC) of the participants. There was a significant difference in the changes for the visual learning test, visual working memory test, and vocabulary (MoCA-K) between the SM70EE and placebo groups after 12 weeks. * *p* < 0.05, ** *p* < 0.01. Abbreviations: SM70EE, *Spirulina maxima* 70% ethanol extract; MoCA-K, Korean version of the Montreal Cognitive Assessment.

**Table 1 nutrients-14-03714-t001:** Demographics of study participants who were placed through random selection into placebo or SM70EE groups.

Variables	SM70EE (Mean ± SD)	Placebo (Mean ± SD)	Total (*n* = 80)	*p* Value ^1^
Sex (male/female)	12/28	9/31	21/59	0.446
Education (years)	10.58 ± 3.66	10.05 ± 3.86	10.31 ± 3.75	0.534
Age (years)	67.68 ± 4.43	68.85 ± 4.89	68.26 ± 4.68	0.264
Alcohol drinking (unit/week)	2.52 ± 5.92	1.30 ± 3.89	1.91 ± 5.02	0.280
Alcohol drinking history (yes/no)	15/25	13/27	28/52	0.639
Smoking (unit/day)	0.25 ± 1.58	0.00 ± 0.00	0.13 ± 1.12	0.323
Smoking history (yes/no)	1/39	0/40	1/79	0.314
Systolic blood pressure (mmHg)	131.50 ± 13.59	134.78 ± 13.03	133.14 ± 13.33	0.275
Diastolic blood pressure (mmHg)	78.03 ± 10.40	80.50 ± 9.01	79.26 ± 9.75	0.259
Heart rate (beats/min)	71.20 ± 8.35	74.40 ± 9.09	72.80 ± 8.82	0.105
Height (cm)	157.73 ± 8.33	157.18 ± 7.69	157.45 ± 7.97	0.760
Weight (kg)	63.19 ± 10.67	63.60 ± 10.13	63.39 ± 10.34	0.861
Body mass index (kg/m^2^)	25.32 ± 3.11	25.67 ± 2.94	25.49 ± 3.01	0.607
K-MMSE	26.38 ± 1.15	26.38 ± 0.98	26.38 ± 1.06	1.000

Values are presented as mean ± standard deviation (SD) or number (percentage). Abbreviations: K-MMSE, Korean mini-mental status examination; SM70EE, *Spirulina maxima* 70% ethanol extract. ^1^ Analyzed using an independent *t*-test.

**Table 2 nutrients-14-03714-t002:** Evaluation of compliance among the participants.

Variables	SM70EE (*n* = 37)	Placebo (*n* = 37)	Total (*n* = 74)	*p* Value ^1^
Prescriptions	253.35 ± 7.62	252.14 ± 7.01	252.74 ± 7.30	0.477
Total intake	234.07 ± 20.05	235.61 ± 15.93	234.84 ± 18.00	0.715
Compliance (%)	92.41 ± 8.15	93.43 ± 6.27	92.92 ± 7.24	0.545

Compliance was calculated by total intake over prescriptions. Values are presented as mean ± standard deviation. Abbreviations: SM70EE, *Spirulina maxima* 70% ethanol extract. ^1^ Analyzed using an independent *t*-test.

**Table 3 nutrients-14-03714-t003:** Measurement of computerized neurocognitive test (CNT) variables at baseline and after the 12-week intervention period.

Variables	SM70EE (Mean ± SD)				Placebo (Mean ± SD)				*p* ^2^	*p* ^3^
	Baseline	12 Weeks	Change Value	*p* ^1^	Baseline	12 Weeks	Change Value	*p* ^1^		
Visual learning test A1	9.77 ± 1.61	10.37 ± 2.30	0.78 ± 1.85	0.130	9.28 ± 1.41	9.75 ± 2.22	0.68 ± 1.42	0.173	0.792	0.242
Visual learning test A2	10.06 ± 1.64	11.00 ± 1.60	0.94 ± 1.54	0.001	10.15 ± 1.50	10.56 ± 1.26	0.38 ± 1.86	0.225	0.177	0.169
Visual learning test A3	10.44 ± 1.86	11.14 ± 1.44	0.69 ± 1.79	0.026	10.56 ± 1.67	10.26 ± 1.44	−0.27 ± 1.82	0.373	0.025	0.007
Visual learning test A4	10.94 ± 1.43	11.14 ± 1.59	0.20 ± 1.39	0.400	11.11 ± 1.75	10.97 ± 1.81	−0.14 ± 1.81	0.653	0.384	0.460
Visual learning test A5	11.06 ± 1.46	11.15 ± 1.50	0.09 ± 1.51	0.731	11.27 ± 1.10	11.18 ± 1.49	−0.08 ± 1.30	0.707	0.610	0.796
Visual learning test (recognition)	10.66 ± 1.11	10.80 ± 1.84	0.14 ± 1.56	0.590	10.26 ± 1.48	10.53 ± 1.52	0.24 ± 1.75	0.405	0.798	0.844
Visual working memory test (accuracy)	26.77 ± 14.85	41.51 ± 11.03	14.75 ± 17.42	0.000	29.31 ± 14.11	33.53 ± 18.78	3.87 ± 18.85	0.220	0.015	0.024
Visual working memory test (corrected accuracy)	20.15 ± 14.12	32.70 ± 15.47	12.55 ± 18.09	0.000	23.29 ± 15.90	26.86 ± 17.09	3.38 ± 17.27	0.242	0.032	0.064
Visual working memory test (reaction time)	636.14 ± 81.08	634.16 ± 71.83	−1.98 ± 86.99	0.895	617.25 ± 80.14	604.70 ± 84.77	−11.87 ± 92.04	0.438	0.644	0.205
Verbal learning test A1	4.59 ± 1.52	5.91 ± 1.64	1.32 ± 1.90	0.000	5.03 ± 1.47	6.55 ± 1.70	1.35 ± 2.06	0.000	0.953	0.191
Verbal learning test A2	7.53 ± 1.81	8.12 ± 2.01	0.59 ± 1.94	0.086	7.36 ± 1.62	8.79 ± 2.00	1.27 ± 1.79	0.000	0.128	0.080
Verbal learning test A3	8.92 ± 1.95	9.25 ± 2.14	0.33 ± 1.84	0.283	9.12 ± 2.04	9.79 ± 1.86	0.62 ± 2.06	0.075	0.530	0.294
Verbal learning test A4	9.84 ± 2.03	10.06 ± 1.69	0.23 ± 1.98	0.530	9.79 ± 2.10	10.85 ± 2.19	0.97 ± 1.96	0.005	0.124	0.061
Verbal learning test A5	10.11 ± 1.51	10.91 ± 2.32	0.80 ± 1.62	0.006	10.23 ± 2.16	11.26 ± 2.12	0.97 ± 1.82	0.002	0.672	0.534
Verbal learning test B	4.57 ± 1.46	4.57 ± 1.42	0.00 ± 1.31	1.000	4.35 ± 1.45	4.12 ± 1.43	−0.22 ± 1.73	0.454	0.551	0.251
Verbal learning test A6	7.91 ± 1.91	8.79 ± 2.59	0.88 ± 1.72	0.005	8.40 ± 2.72	9.46 ± 2.78	1.00 ± 1.96	0.004	0.789	0.569
VLT A20 (delayed recall)	7.19 ± 3.11	8.81 ± 2.93	1.61 ± 2.45	0.000	7.06 ± 2.33	8.79 ± 3.04	1.54 ± 2.44	0.000	0.902	0.898
VLT REC (delayed recognition)	11.28 ± 1.54	12.25 ± 1.83	0.97 ± 1.78	0.002	10.83 ± 2.05	12.08 ± 1.78	1.22 ± 1.87	0.000	0.570	0.900
VLT A1A5 total (trial-learning)	41.27 ± 8.09	58.70 ± 16.02	17.42 ± 14.86	0.000	42.44 ± 9.97	64.36 ± 22.37	21.32 ± 20.95	0.000	0.369	0.289
VLT A1A5 total (average)	8.25 ± 1.62	11.74 ± 3.20	3.48 ± 2.97	0.000	8.49 ± 1.99	12.87 ± 4.47	4.26 ± 4.19	0.000	0.369	0.289
VLT learning slope A5A1 (Learning Indicator)	5.33 ± 1.41	4.82 ± 1.67	−0.52 ± 1.79	0.108	4.71 ± 1.86	4.31 ± 2.08	−0.38 ± 2.13	0.287	0.773	0.581
VLT A5 A20 (memory retention)	3.17 ± 1.89	2.42 ± 2.12	−0.75 ± 2.32	0.061	3.55 ± 1.25	2.30 ± 1.61	−1.11 ± 1.61	0.000	0.446	0.548

Values are presented as mean ± standard deviation (SD). Abbreviations: SM70EE, *Spirulina maxima* 70% ethanol extract. ^1^ Analyzed using a paired *t*-test between baseline and 12 weeks in each group. ^2^ Analyzed using an independent *t*-test for changes in values between the groups. ^3^ Analyzed using ANCOVA (baseline as a covariate). Abbreviations: VLT, verbal learning test; REC, recognition.

**Table 4 nutrients-14-03714-t004:** Changes in the Korean version of the Montreal Cognitive Assessment (MoCA-K) test before and after 12 weeks of SM70EE product consumption (excluding outliers).

Variables	SM70EE (Mean ± SD)				Placebo (Mean ± SD)				*p* ^2^	*p* ^3^
	Baseline	12 Weeks	Change Value	*p* ^1^	Baseline	12 Weeks	Change Value	*p* ^1^		
Visuospatial/executive	3.52 ± 0.83	3.76 ± 0.61	0.14 ± 1.00	0.147	3.56 ± 0.80	3.63 ± 0.75	0.14 ± 1.09	0.773	1.000	0.446
Vocabulary	2.78 ± 0.42	2.84 ± 0.37	0.17 ± 0.38	0.423	2.47 ± 0.74	2.44 ± 0.69	0.06 ± 0.63	0.812	0.354	0.043
Attention	5.22 ± 0.72	5.42 ± 0.73	0.28 ± 0.74	0.165	5.00 ± 0.95	5.44 ± 0.66	0.38 ± 0.92	0.023	0.602	0.676
Language	2.69 ± 0.47	2.74 ± 0.44	0.00 ± 0.00	0.487	2.53 ± 0.66	2.76 ± 0.43	0.03 ± 0.67	0.103	0.801	0.714
Abstraction	1.71 ± 0.46	2.00 ± 0.00	0.29 ± 0.46	0.001	1.61 ± 0.50	2.00 ± 0.00	0.36 ± 0.49	0.000	0.504	-
Delayed recall	0.66 ± 1.03	1.54 ± 1.79	0.67 ± 1.53	0.003	0.82 ± 1.10	1.33 ± 1.36	0.54 ± 1.04	0.004	0.691	0.300
Orientation	5.72 ± 0.45	5.75 ± 0.44	0.18 ± 0.39	0.744	5.68 ± 0.47	5.71 ± 0.46	0.03 ± 0.57	0.768	0.202	0.767
Total	22.52 ± 2.14	23.88 ± 2.39	1.58 ± 2.25	0.001	21.69 ± 2.64	23.22 ± 2.93	1.49 ± 2.44	0.003	0.861	0.746

Values are presented as mean ± standard deviation (SD). Abbreviations: SM70EE, *Spirulina maxima* 70% ethanol extract. ^1^ Analyzed using a paired *t*-test between baseline and 12 weeks in each group. ^2^ Analyzed using an independent *t*-test for changes in values between the groups. ^3^ Analyzed using an analysis of covariance (baseline as a covariate).

**Table 5 nutrients-14-03714-t005:** Changes in biomarkers before and after 12 weeks of SM70EE product consumption.

Variables	SM70EE				Placebo				*p* ^2^	*p* ^3^
	Baseline	12 Weeks	Change Value	*p* ^1^	Baseline	12 Weeks	Change Value	*p* ^1^		
BDNF (pg/mL)	23499.19 ± 4976.21	21352.48 ± 4461	−2236.5 ± 3359.17	0.003	26759.21 ± 5458.02	23599.93 ± 5182.22	−3500.73 ± 3319.6	0.000	0.118	0.945
TAC (mmol/L)	0.93 ± 0.15	0.93 ± 0.16	0.01 ± 0.09	0.990	0.93 ± 0.18	0.86 ± 0.17	−0.02 ± 0.13	0.059	0.297	0.066
Amyloid β (1–40) (pg/mL)	279.05 ± 55.64	275.97 ± 45.37	4.32 ± 25.15	0.612	261.78 ± 45.87	268.62 ± 54.35	6.28 ± 24.88	0.170	0.745	0.412
Amyloid-β (1–42) (pg/mL)	2.95 ± 1.52	2.76 ± 1.24	−0.27 ± 1.15	0.292	3.77 ± 3.24	3.5 ± 2.46	−0.04 ± 1.17	0.373	0.401	0.413
Amyloid-β (40/42)	102.46 ± 49.93	107.8 ± 52.05	5.34 ± 40.6	0.435	90.86 ± 42.47	87.84 ± 38.56	3.66 ± 28.67	0.986	0.841	0.224

Values are presented as mean ± SD. Abbreviations: BDNF, brain-derived neurotrophic factor; TAC, total antioxidant capacity; SM70EE, *Spirulina maxima* 70% ethanol extract. ^1^ Analyzed using a paired *t*-test between baseline and 12 weeks in each group. ^2^ Analyzed using an independent *t*-test for changes in values between the groups. ^3^ Analyzed using an analysis of covariance (baseline as a covariate.

**Table 6 nutrients-14-03714-t006:** Frequency of occurrence of adverse events among the participants.

Variables	SM70EE (*n* = 40)	Placebo (*n* = 40)	Total (*n* = 80)	*p* Value ^1^
Frequency of adverse events (%)	1(2.5)	0(0)	1(1.25)	1.000

Values are presented as mean ± standard deviation (SD). Abbreviations: SM70EE, *Spirulina maxima* 70% ethanol extract. ^1^ Analyzed using a Chi-square test or Fisher’s exact test.

**Table 7 nutrients-14-03714-t007:** Changes in the diagnostic medical test before and after 12 weeks of SM70EE product consumption.

Variables	Normal Range	SM70EE (Mean ± SD)				Placebo (Mean ± SD)				*p* ^2^
		Baseline	12 Weeks	Change Value	*p* ^1^	Baseline	12 Weeks	Change Value	*p* ^1^	
WBC (×10^3^/μL)	4.8~10.8	5.72 ± 1.16	5.63 ± 1.16	−0.09 ± 0.91	0.554	5.54 ± 1.19	5.54 ± 1.16	0.00 ± 1.22	0.979	0.705
RBC (×100^3^/μL)	4.2~5.4	4.48 ± 0.32	4.40 ± 0.32	−0.07 ± 0.17	0.015	4.56 ± 0.39	4.51 ± 0.34	−0.05 ± 0.18	0.075	0.665
Hb (g/dL)	12~16	13.75 ± 1.17	13.53 ± 1.14	−0.22 ± 0.54	0.016	13.93 ± 1.33	13.72 ± 1.15	−0.21 ± 0.61	0.035	0.936
Hct (%)	37~47	40.97 ± 3.44	39.97 ± 3.11	−1.01 ± 1.48	0.000	41.51 ± 3.55	40.54 ± 2.93	−0.98 ± 1.78	0.001	0.940
Platelet (×10^3^/μL)	130~450	236.77 ± 41.86	230.64 ± 38.44	−6.13 ± 22.08	0.091	244.68 ± 45.05	247.13 ± 48.39	2.45 ± 26.30	0.559	0.121
ALP (IU/L)	45~129	70.18 ± 16.31	69.90 ± 16.10	−0.28 ± 8.02	0.827	70.58 ± 16.43	69.95 ± 16.18	−0.63 ± 7.34	0.593	0.843
Gamma GT (IU/L)	8~48	22.49 ± 15.84	25.56 ± 20.74	3.08 ± 11.26	0.096	28.75 ± 23.65	28.58 ± 22.03	−0.18 ± 9.75	0.910	0.174
AST (IU/L)	12~33	25.51 ± 5.92	23.97 ± 4.81	−1.54 ± 5.12	0.068	28.33 ± 10.99	24.93 ± 6.36	−3.40 ± 10.17	0.041	0.310
ALT (IU/L)	5~35	23.95 ± 8.73	22.62 ± 8.18	−1.33 ± 6.53	0.210	28.00 ± 14.17	24.83 ± 10.48	−3.18 ± 11.37	0.085	0.382
Total bilirubin (mg/dL)	0.2~1.2	0.83 ± 0.20	0.83 ± 0.23	0.00 ± 0.17	0.861	0.90 ± 0.38	0.87 ± 0.36	−0.03 ± 0.30	0.528	0.527
Total protein (g/dL)	6.7~8.3	7.43 ± 0.35	7.14 ± 0.41	−0.28 ± 0.37	0.000	7.43 ± 0.35	7.25 ± 0.41	−0.18 ± 0.37	0.004	0.210
Albumin (g/dL)	3.5~5.3	4.39 ± 0.22	4.30 ± 0.21	−0.09 ± 0.18	0.002	4.45 ± 0.20	4.38 ± 0.19	−0.08 ± 0.20	0.024	0.644
BUN (mg/dL)	8~23	16.00 ± 4.26	16.33 ± 4.68	0.33 ± 3.36	0.539	15.30 ± 3.75	16.08 ± 4.82	0.78 ± 3.99	0.227	0.597
Creatinine (mg/dL)	0.7~1.7	0.70 ± 0.18	0.71 ± 0.20	0.01 ± 0.08	0.305	0.67 ± 0.19	0.70 ± 0.21	0.03 ± 0.09	0.073	0.479
Cholesterol (mg/dL)	~200	197.97 ± 33.26	189.54 ± 34.37	−8.44 ± 22.23	0.023	198.78 ± 33.03	191.03 ± 37.	−7.75 ± 26.91	0.076	0.902
Triglyceride (mg/dL)	~200	127.90 ± 49.97	136.31 ± 78.92	8.41 ± 58.02	0.371	145.60 ± 66.77	144.15 ± 70.12	−1.45 ± 53.21	0.864	0.433
HDL Cholesterol (mg/dL)	48.9~73.5	53.67 ± 11.18	52.28 ± 12.20	−1.38 ± 7.30	0.244	54.35 ± 14.66	52.38 ± 14.63	−1.98 ± 8.59	0.154	0.743
LDL C (mg/dL)	0~140	113.72 ± 24.37	110.00 ± 26.85	−3.72 ± 19.77	0.248	112.63 ± 27.59	110.78 ± 32.58	−1.85 ± 22.88	0.612	0.699
Glucose (mg/dL)	74~106	95.92 ± 20.65	95.31 ± 22.78	−0.62 ± 9.73	0.695	98.63 ± 18.61	94.95 ± 17.85	−3.68 ± 11.81	0.056	0.213
CK (IU/L)	50~200	114.59 ± 80.54	121.69 ± 72.51	7.10 ± 78.14	0.574	89.73 ± 32.04	100.40 ± 58.73	10.68 ± 46.15	0.152	0.805
LD (IU/L)	218~472	387.90 ± 62.52	405.69 ± 59.86	17.79 ± 50.46	0.034	405.43 ± 67.57	403.23 ± 70.24	−2.20 ± 58.19	0.812	0.107
SG	1.005~1.030	1.02 ± 0.01	1.02 ± 0.01	0.00 ± 0.01	0.527	1.02 ± 0.00	1.02 ± 0.01	0.00 ± 0.01	0.853	0.774
PH	4.5 ~9.0	6.23 ± 0.68	6.15 ± 0.81	−0.08 ± 0.62	0.446	6.26 ± 0.82	6.19 ± 0.99	−0.08 ± 0.66	0.474	0.989

Values are presented as mean ± standard deviation (SD). Abbreviations: ALP, alkaline phosphatase; ALT, alanine aminotransferase; AST, aspartate aminotransferase; BUN, blood urea nitrogen; C, cholesterol; CK, creatine kinase; Hb, hemoglobin; Hct, hematocrit; HDL, high-density lipoprotein, LD, lactate dehydrogenase; LDL, low-density lipoprotein; SM70EE, *Spirulina maxima* 70% ethanol extract; RBC, red blood count; WBC, white blood count. ^1^ Analyzed using a paired *t*-test between baseline and 12 weeks in each group. ^2^ Analyzed using an independent *t*-test for changes in values between the groups.

**Table 8 nutrients-14-03714-t008:** Changes in test results concerning vital signs before and after 12 weeks of SM70EE product consumption.

Variables	SM70EE (Mean ± SD)				Placebo (Mean ± SD)				*p* ^2^
	Baseline	12 Weeks	Change Value	*p* ^1^	Baseline	12 Weeks	Change Value	*p* ^1^	
Systolic blood pressure (mmHg)	128.00 ± 16.00	126.63 ± 13.64	−1.38 ± 14.30	0.547	125.93 ± 14.29	126.30 ± 15.01	0.38 ± 12.79	0.854	0.566
Diastolic blood pressure (mmHg)	76.13 ± 13.20	73.85 ± 10.12	−2.28 ± 12.65	0.262	75.20 ± 8.59	75.70 ± 9.96	0.50 ± 9.08	0.729	0.263
Heart rate (beats/min)	76.45 ± 8.95	72.98 ± 8.77	−3.48 ± 10.20	0.037	78.63 ± 10.18	76.63 ± 9.49	−2.00 ± 9.38	0.185	0.503

Values are presented as mean ± standard deviation (SD). Abbreviations: SM70EE, *Spirulina maxima* 70% ethanol extract. ^1^ Analyzed using a paired *t*-test between baseline and 12 weeks in each group. ^2^ Analyzed using an independent *t*-test for changes in values between the groups.

**Table 9 nutrients-14-03714-t009:** Changes in body mass index (BMI) and body weight results before and after 12 weeks of SM70EE product consumption.

Variables	SM70EE (Mean ± SD)				Placebo (Mean ± SD)				*p* ^2^
	Baseline	12 Weeks	Change Value	*p* ^1^	Baseline	12 Weeks	Change Value	*p* ^1^	
Body weight (kg)	63.19 ± 10.67	63.23 ± 11.10	0.05 ± 1.53	0.853	63.60 ± 10.13	63.39 ± 9.91	−0.21 ± 1.47	0.373	0.449
BMI (kg/m^2^)	25.32 ± 3.11	25.35 ± 3.41	0.03 ± 0.62	0.761	25.67 ± 2.94	25.58 ± 2.92	−0.09 ± 0.56	0.340	0.385

Values are presented as mean ± standard deviation (SD). Abbreviations: SM70EE, *Spirulina maxima* 70% ethanol extract. ^1^ Analyzed using a paired *t*-test between baseline and 12 weeks in each group. ^2^ Analyzed using an independent *t*-test for changes in values between the groups.

**Table 10 nutrients-14-03714-t010:** Changes in metabolic equivalent of task (MET) results before and after 12 weeks of SM70EE product consumption.

Variables	SM70EE (Mean ± SD)				Placebo (Mean ± SD)				*p* ^2^
	Baseline	12 Weeks	Change Value	*p* ^1^	Baseline	12 Weeks	Change Value	*p* ^1^	
MET value (min/week)	3506.49 ± 3772.38	3830.27 ± 3912.30	323.78 ± 5273.81	0.711	2644.32 ± 2829.78	2594.16 ± 2685.63	−50.16 ± 2569.40	0.906	0.700

Values are presented as mean ± standard deviation (SD). Abbreviations: SM70EE, *Spirulina maxima* 70% ethanol extract. ^1^ Analyzed using a paired *t*-test between baseline and 12 weeks in each group. ^2^ Analyzed using an independent *t*-test for changes in values between the groups.

## Data Availability

The data presented in this study are available in the article.

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
