# Peer review of "The Effects of Spirulina maxima Extract on Memory Improvement in Those with Mild Cognitive Impairment: A Randomized, Double-Blind, Placebo-Controlled Clinical Trial"

_nutrients, 2022, doi:10.3390/nu14183714_

Round 1

Reviewer 1 Report

The manuscript ¨ The Effects of Spirulina maxima Extract on Memory Improvement in Those with Mild Cognitive Impairment: A Randomized, Double-Blind, Placebo-Controlled Clinical Trial ¨ by Woon-Yong Choi et al. presents the results of a clinical trial about the effects of the Spirulina maxima 70% ethanol extract (SM70EE) to the memory function.

The manuscript is well organized. The content is supported by appropriate references and statistical analyses.

There are a few questions and suggestions to the authors.

11.     Why were the patients followed only in the 12 weeks period? Do you think that this period is long enough to show progression in memory function? 

22.      In the abstract (P 1, line 22) you said: “The participants were examined at baseline and after a 12-week interval...“. Further, in the Clinical trial section (P 3, line 147-151) there is: “Subsequently, every 6 weeks, the participants visited the Functional Food Clinical Trial Support Center at Cheonbuk National University Hospital and underwent tests related to memory improvement, vital signs, drug administration, changes in medical conditions, adverse reactions, and tests specified in the clinical test plan. “. According to this, you should have the data for the 6 week period as well. It would be interesting to see if any changes were present at that time. Is there a reason why you didn’t show those results?

33.      How did you determine that 1 g/day of SM70EE is the best dose for trial and why it was not related to the body mass of participants? Can you explain why you didn’t try with some other doses?

44.      Abstract: Is 2nd sentence (P1, L 16-17) related to this clinical trial? If it is, then you should write “through THIS clinical trial”. This same sentence would be better somewhere at the end of the abstract in combination with the sentence in L 28-29. In L 26 add “improvement IN visual”

55.      Introduction: The sentence on P2, L 70-72 has several problems. You used term strongly effective which is, in my opinion, too strong for the results and P values that you presented in results section, and it should be just effective. Further, in line 72 you wrote “as shown in previous our studies [15, 16]”. Those references are wrong, I think that there should be 6 and 7. Additionally, you should change the word order (as shown in our previous studies) and add that those were studies performed on animals.

66.      In Materials and Methods section I didn’t find description of the methods used for BDNF, Aβ levels and TAC.

77.      In the lowest boxes of Fig 1 should be written “Included in primary analyses (or analysis?)”. Please correct.

88.      Caption for Figure 1 (P 4, L 157-161) is the same as for Figure 2. Replace it with an adequate description of Fig 1.

99.      Correct the caption of Figure 2 (P 9, L 307-3011): “Changes OF values”, “There was a SIGNIFICANT difference”.

110.  P 10, L 319: consumption (remove -).

111.  The last section in the Discussion consists of the sentences that are already mentioned in the discussion and are the same: L 433-435 same as L 350-352; L 436-440 same as 359-364; L 442-448 same as 364-371. This section should be rewritten or excluded.

112.  There are several incorrect expressions in the Conclusions: L 453 – TO the CNT; L 456 – vocabulary SHOWED a; L 457 – to be safe FOR the…

Reviewer 2 Report

My suggestions:

1. In the introduction I would discuss a little more about Spirulina and its benefits against different diseases beyond AD.

2. Did the individuals/patients have any history of diabetes or high blood sugar level? If yes, it would be interesting to know. 

3. In the discussion, I would add a pathway figure, which may show, how SM70EE would show neuroprotective effects in the case of MCI patients. 

4. Is it possible that SM70EE is slowing down the disease progression in case of AD or other forms of neurodegeneration too?

5. Are there any side effects of SM70EE exposure?

Round 2

Reviewer 2 Report

The authors fulfilled my suggestions. Thank you,.